# Effectiveness and Safety of Probiotics for Patients with Constipation-Predominant Irritable Bowel Syndrome: A Systematic Review and Meta-Analysis of 10 Randomized Controlled Trials

**DOI:** 10.3390/nu14122482

**Published:** 2022-06-15

**Authors:** Xue Shang, Fen-Fen E, Kang-Le Guo, Yan-Fei Li, Hong-Lin Zhao, Yan Wang, Nan Chen, Tao Nian, Chao-Qun Yang, Ke-Hu Yang, Xiu-Xia Li

**Affiliations:** 1Health Technology Assessment Center, Evidence-Based Social Science Research Center, School of Public Health, Lanzhou University, Lanzhou 730000, China; shangx20@lzu.edu.cn (X.S.); eff20@lzu.edu.cn (F.-F.E.); guokl19@lzu.edu.cn (K.-L.G.); ywang2020@lzu.edu.cn (Y.W.); chenn19@lzu.edu.cn (N.C.); niant21@lzu.edu.cn (T.N.); yangchq2020@lzu.edu.cn (C.-Q.Y.); 2Evidence Based Medicine Center, School of Basic Medical Sciences, Lanzhou University, Lanzhou 730000, China; yfli2021@lzu.edu.cn; 3Key Laboratory of Evidence Based Medicine and Knowledge Translation of Gansu Province, Lanzhou 730000, China; 4Institute of Social Medicine and Health Management, School of Public Health, Lanzhou University, Lanzhou 730000, China; zhaohl20@lzu.edu.cn

**Keywords:** irritable bowel syndrome, constipation, probiotics, meta-analysis, systematic review

## Abstract

To perform a systematic review and meta-analysis to evaluate the effectiveness and safety of probiotics in the treatment of constipation-predominant irritable bowel syndrome (IBS-C), we searched for randomized controlled trials (RCTs) comparing probiotic care versus placebos for patients with IBS-C in five comprehensive databases (March 2022). The risk of bias was assessed using the Cochrane Collaboration Risk of Bias Tool. RevMan 5.3 was used to perform a meta-analysis on stool consistency, abdominal pain, bloating, quality of life (QoL), fecal *Bifidobacterium* and *Lactobacillus* counts, and adverse events. The GRADE approach was used to evaluate the certainty of the evidence. Ten RCTs involving 757 patients were included. Only three studies were rated as having a low risk of bias. The meta-analysis results show that, compared to the placebo, probiotics significantly improved stool consistency (MD = 0.72, 95% CI (0.18, 1.26), *p* < 0.05, low quality) and increased the number of fecal *Bifidobacteri**a* (MD = 1.75, 95% CI (1.51, 2.00), *p* < 0.05, low quality) and *Lactobacillus* (MD = 1.69, 95% CI (1.48, 1.89), *p* < 0.05, low quality), while no significant differences were found in abdominal pain scores, bloating scores, QoL scores, or the incidence of adverse events (*p* > 0.05). The low-to-very low certainty evidence suggests that probiotics might improve the stool consistency of patients with IBS-C and increase the number of *Bifidobacteria* and *Lactobacilli* in feces with good safety. However, more high-quality studies with large samples are needed to verify the findings.

## 1. Introduction

As a pervasive functional bowel disease, irritable bowel syndrome (IBS) is characterized by abdominal pain and changes in defecation habits, principally manifested as diarrhea (IBS-D), constipation (IBS-C), or both (IBS-M) [1]. Currently, the global preponderance of IBS is about 11% [2]. Approximately one-third of the patients with IBS experience constipation as their predominant bowel symptom (IBS with constipation (IBS-C) subtype) [3]. IBS-C is defined by the Rome IV criteria as recurrent abdominal pain associated with defecation and/or a change in stool frequency or form [4]. Women with IBS are more than twice as likely to meet the criteria for IBS-C as men with IBS [5]. In a United States (US) population-based survey of 1667 individuals meeting the symptom criteria for IBS-C, the most common symptoms reported by patients were abdominal pain (83%), bloating (78%), and straining (75%) [6]. Alterations in the microbial composition of the feces and small intestine have also been found in IBS patients [7]. Some studies have reported that the number of *Bifidobacteria* and *Lactobacilli* in the feces of IBS patients was less than that of normal subjects, whereas other studies reported that the number was greater than that of normal subjects [8,9,10,11,12,13,14]. Existing evidence revealed that IBS is a multifactorial disease, and the exact etiology and pathogenesis of IBS are not completely clear. Yet, gastrointestinal flora disorder, mild mucosal inflammation, chronic immune activation, changes in intestinal permeability, and psychosocial factors are implicated in IBS [4,15,16,17].

Patients with IBS-C are often recommended and use lifestyle modifications or medications, such as nonprescription fiber laxatives or stool softeners, to manage their symptoms [18]. However, these therapies can only be palliative and supportive for specific symptoms, do not address the issue of long-term efficacy, and are often associated with patient dissatisfaction, potentially resulting in the use of multiple therapies, as well as the repeated switching of medications [18,19]. Meanwhile, grievous adverse drug reactions also affect patients’ compliance with treatment. A questionnaire survey was conducted on 94 patients with IBS, and only 12/94 (12.8%) were satisfied with the treatment they had been given [20,21]. The failure of existing treatment options to adequately control the symptoms of IBS-C has been shown to result in increased treatment costs and healthcare resource use [22]. Based on the above, it is necessary to investigate alternative therapies.

When properly ingested, probiotics are living microorganisms that have potential health benefits [23,24]. The effectiveness of probiotics in improving IBS symptoms has been tentatively recognized, and several probiotic strains have been demonstrated to benefit patients with IBS [25,26,27]. For example, supplementation with *Bifidobacterium* was shown to regulate immune function, intestinal microbiota, and gut mucosal adhesion in patients with IBS [28]. Compared to pharmacologic therapy, probiotic treatment has few adverse events, favorable efficacy, good tolerance, and addresses long-term effects. It also significantly enhances patient compliance with probiotic therapy. Thus, probiotics may be a potential treatment for IBS-C. However, greater evidence based on a systematic review is needed.

Based on the international consensus and existing evidence, probiotics are beneficial in improving overall symptoms in patients with IBS [29,30,31,32,33,34,35], but there are still contradictory reports on improvements in specific symptoms [36,37,38]. In addition, there is weak evidence on the efficacy of probiotics in patients with IBS-C. Therefore, we conducted a systematic review and meta-analysis to investigate the effectiveness and safety of probiotics in IBS-C patients.

## 2. Methods

### 2.1. Inclusion and Exclusion Criteria

Any randomized controlled trial (RCT) of probiotics in the treatment of constipation-predominant irritable bowel syndrome that conformed to the random allocation principles of trial design were included, including parallel and crossover designs. The specific literature inclusion criteria were as follows:Population characteristics, IBS-C patients diagnosed with specific criteria (Rome I, II, III, IV), not limited by age, race, or gender [39,40,41,42];For intervention factors, the treatment group received probiotics in any form (i.e., tablet, powder, capsule, soft gel, or fortified food forms), species, strains, dose, or treatment regimen, while the control group received a placebo;The primary outcomes included stool consistency and changes in the number of fecal colonies. The secondary outcomes were abdominal pain, bloating, QoL, and adverse events.

Studies that met the following criteria were excluded:Animal trials;Duplicate studies;Studies with incomplete data (e.g., abstracts, conference proceedings, or research protocols, among others). The authors were contacted when data were not available.

### 2.2. Literature Search

Five electronic databases, PubMed, Embase, the Cochrane Library, Web of Science, and China Biology Medicine (CBM), were comprehensively searched from inception to March 2021, and the search was updated in March 2022. In addition, we conducted a supplementary inquiry of gray literature and the relevant references. The Medical Subject Headings (MeSH) search terms, Boolean operators and truncation symbols (*) in combinations with key words were as follows: (irritable bowel syndrom* OR IBS-C OR constipat*) AND (probiotic* OR Bacillus[MeSH] OR Bifidobacterium[MeSH] OR Lactobacillus[MeSH] OR Streptococcus[MeSH] OR Enterococcus[MeSH] OR Propionibacterium[MeSH] OR Saccharomyces[MeSH] OR Clostridium[MeSH]) AND (random* OR controlled clinical trial* OR blind* OR single-blind* OR double-blind* OR trebleblind* OR triple-blind* OR RCT).

### 2.3. Literature Screening and Data Extraction

Literature screening and data extraction were performed independently by two reviewers. EndNote X9 software (Clarivate Analytics; Philadelphia, PA, USA) was used to reject duplicate articles. Subsequently, two reviewers screened the titles and abstracts according to the inclusion criteria and then read the full-text articles to confirm whether the studies met the criteria for inclusion. Any disagreements were resolved by discussions among the two reviewers or by consultations with the third reviewer.

A predesigned table was utilized to extract the data as follows:Basic information on the included RCTs, including the year of publication, country, and author(s);The main characteristics of participants, including the sample size, age, gender, and the diagnostic criteria;Details of the interventions implemented (e.g., probiotic strains, dose, duration, form) and the use of placebo;Qualitative and quantitative analysis results of outcome indicators (e.g., the report of different symptoms and the use of scales).

One reviewer independently extracted data from each article, and a second reviewer checked the content.

### 2.4. Assessment of Risk of Bias

Two authors independently assessed the risk of bias of the included RCTs according to the tool recommended by the Cochrane Handbook V.5.1.0 (Cochrane Collaboration; London, UK) [43]. Seven domains were considered, including random sequence generation, allocation concealment, the blinding of participants and personnel, blinding of the outcome assessment, incomplete outcome data, selective reporting, and other biases. If an RCT received yes endorsements across all domains, the study was considered to be at low risk of bias. Alternatively, if one or more domains were flagged with a no, the study was considered to be at high risk of bias. Otherwise, the study’s risk of bias was deemed unclear [44]. Any disagreement was resolved by a third investigator.

### 2.5. Data Synthesis and Statistical Analysis

The meta-analysis was performed using RevMan version 5.3 software (The Nordic Cochrane Centre; Copenhagen, Denmark) [45]. For a specific outcome assessed by the same measures (scale), the continuous variable was synthesized using the mean difference (MD) with a 95% confidence interval (CI). Inversely, standardized mean difference (SMD) and 95% CI were used when the measured scales were different. The odds ratio (OR) with 95% CI was used to represent dichotomous variables. Chi- (*χ*^2^) and I-squared (*I*^2^) tests were used to evaluate statistical heterogeneity, and when *p* < 0.05 and *I*^2^ > 50% signified high heterogeneity, the variable was entered in a random-effects model. Otherwise, a fixed-effects model was selected [46]. Finally, funnel plots were generated to detect publication bias.

### 2.6. Certainty Assessment

We used the Grades of Recommendation, Assessment, Development, and Evaluation (GRADE) system to assess the certainty of the evidence associated with specific outcomes and constructed a table summarizing the evidence findings. The certainty level started at a pre-specified level from the study design (high certainty for RCTs). Five downgrading factors, i.e., study limitations, inconsistency (high statistical heterogeneity), indirectness, imprecision, and publication bias, were then used to estimate the evidence quality for each outcome. Each factor was judged as not serious, serious, or very serious. Correspondingly, according to the judgment, the outcomes were divided into no downgrade (not serious), downgrade by one level (serious), or downgrade by two levels (very serious). In the end, each outcome was judged as different levels of evidence, including high, moderate, low, and very low [47,48].

## 3. Results

### 3.1. Selection Results for the Included Studies

The explicit literature screening process is illustrated in Figure 1. The preliminary database retrieval and supplemental retrieval yielded a total of 5867 records, and 1283 duplicated records were eliminated. According to the inclusion and exclusion criteria, 4318 records were excluded by reading the titles and abstracts, and the full texts of 266 records were accessed. After reviewing the full texts, 256 reports were excluded due to the subjects not matching (diarrhea-type IBS or unclassified, *n* = 75); irrelevant interventions, which were a no-control or a conventional control group (*n* = 8); probiotics combined with herbal or other drug interventions (*n* = 82); irrelevant study design (*n* = 22); irrelevant results, which did not meet our inclusion criteria (*n* = 14), or incomplete data (*n* = 45). Finally, ten RCTs with 757 patients were included in the meta-analysis.

### 3.2. Characteristics of the Included Studies

The general characteristics of the included trials are presented in Table 1. All included RCTs were published between 2007 and 2021. The sampling size ranged from 11 to 267 patients, and the duration ranged from four weeks to twelve weeks. The majority of the trials were conducted in Europe (*n* = 4), followed by Asia (*n* = 3). The most commonly used probiotics in the included studies were *Lactobacillus* [49,50,51,52,53,54,55,56] and *Bifidobacterium* [49,51,53,54,56], which were implemented alone or in combination with other genera. *Saccharomyces cerevisiae* was used in the yeast trials [57,58]. *Streptococcus thermophilus* and *Lactobacillus* bulgaricus [49,54] have also been studied as yogurt starter cultures. Most studies reported multispecies probiotic treatment rather than single-strain probiotics, with a mixture of up to six probiotic strains used in some preparations [49]. Some strains (such as *L. acidophilus* and *L. rhamnosus*) were used more in the studies, but many strains were used only in a few combinations. In these studies, probiotics were administered to the patients in the form of capsules, fermented milk, and food supplements. Probiotics also varied in dosage, and the daily dose of probiotics was collated by CFU/g or CFU/pot. The main outcome measures of interest concentrated on improvements in abdominal pain, bloating, stool consistency, QoL, adverse events, and changes in the number of fecal colonies.

### 3.3. Risk of Bias of the Included Studies

As shown in Figure 2, three studies were rated as low risk of bias due to complete reporting of information on each item, and seven studies were assessed as unclear risk of bias because of incomplete reporting. There were six RCTs with insufficient information generated by the random sequence [50,52,54,55,56,58], and seven RCTs had imperfect allocation concealment information [50,52,53,54,55,56,58]. Five RCTs had unclear designs and blinding implementation [50,52,54,56,58]; two of them did not report the blinding of researchers and participants, and five of them did not report the blinding of outcome assessments.

### 3.4. Meta-Analysis

#### 3.4.1. Stool Consistency Scores

Three RCTs [50,51,58] involving 71 patients reported the effects of probiotics on the stool consistency scores of IBS-C patients. Overall, the meta-analysis results show that compared to placebos, probiotics improved the stool consistency scores of IBS-C patients (MD = 0.72, 95% CI [0.18, 1.26], *p* < 0.05). In the subgroup analysis, an eight-week duration showed a good effect on stool consistency (MD = 0.71, 95% CI [0.11, 1.32], *p* < 0.05), but twelve weeks showed no effect (MD = 0.75, 95% CI [−0.53, 2.03], *p* > 0.05; Figure 3). 

#### 3.4.2. Fecal Bifidobacterium and Lactobacillus Counts

Two RCTs [55,56] involving 74 patients reported the effects of probiotics on *Bifidobacterium* and *Lactobacillus* fecal counts in IBS-C patients. Overall, the meta-analysis results demonstrate that compared to placebos, probiotics significantly increased the number of fecal *Bifidobacteria* (MD = 1.75, 95% CI [1.51, 2.00], *p* < 0.05) and *Lactobacilli* (MD= 1.69, 95% CI [1.48, 1.89], *p* < 0.05) in IBS-C patients. In the subgroup analysis, a four-week duration showed good effects on *Bifidobacterium* (MD = 1.76, 95% CI [1.51, 2.01], *p* < 0.05) and *Lactobacillus* (MD = 1.69, 95% CI [1.48, 1.90], *p* < 0.05), but an eight-week duration showed no effect on *Bifidobacterium* (MD = −0.15, 95% CI [−4.88, 4.58], *p* > 0.05) or *Lactobacillus* (MD= 0.54, 95% CI [−2.79, 3.87], *p* > 0.05; Figure 4).

#### 3.4.3. Abdominal Pain Scores

Four RCTs [50,54,57,58] involving 488 patients reported the effects of probiotics on abdominal pain scores in IBS-C patients. Overall, the meta-analysis results show that there were no significant differences in abdominal pain scores between the probiotic and placebo groups (SMD = −0.28, 95% CI [−0.60, 0.05], *p* > 0.05; Figure 5a). In the subgroup analysis (Figure 5a), an eight-week duration showed good effects on improving abdominal pain (SMD = −1.28, 95% CI [−2.26, −0.29], *p* < 0.05), but six-week (SMD = −0.14, 95% CI [−0.38, 0.10], *p* > 0.05) and twelve-week (SMD = −0.21, 95% CI [−0.66, 0.24], *p* > 0.05) durations showed no effect. The sensitivity analysis indicated that the results were unstable. When one study was excluded [58], the combined results and heterogeneity changed significantly. The meta-analysis results demonstrate that probiotics significantly reduced abdominal pain scores in IBS-C patients compared to placebos (SMD = −0.20, 95% CI [−0.38, −0.01], *p* < 0.05, *I*^2^ = 0%; Figure 5b).

#### 3.4.4. Bloating Scores

Two RCTs [54,57] involving 447 patients reported the effects of probiotics on bloating scores in IBS-C patients. Overall, the meta-analysis results indicate that there were no significant differences in bloating scores between the probiotic and placebo groups (SMD = −0.14, 95% CI [−0.46, 0.18], *p* > 0.05; Figure 6a). In the subgroup analysis (Figure 6a), a 12-week duration showed good effects on improving bloating (SMD = −0.31, 95% CI [−0.61, −0.02], *p* < 0.05), but a six-week duration showed no effect (SMD = 0.01, 95% CI [−0.23, 0.25], *p* > 0.05). The results were unstable when we excluded one study assessed as having an unclear risk of bias [54]. The results show that probiotics significantly decreased the bloating scores of IBS-C patients (SMD = −0.31, 95% CI [−0.61, −0.02], *p* < 0.05; Figure 6b).

#### 3.4.5. QoL Scores

Three RCTs [50,53,54] involving 487 patients reported the effects of probiotics on QoL scores in IBS-C patients. Overall, the meta-analysis results demonstrate that there was no significant difference in QoL scores between the probiotic and placebo groups (SMD = −3.92, 95% CI [−8.09, 0.25], *p* > 0.05; Figure 7a). In the subgroup analysis, an eight-week duration showed good effects on improving QoL (SMD = −8.06, 95% CI [−8.92, −7.21], *p* < 0.05), but six-week (SMD = −0.05, 95% CI [−0.29, 0.19], *p* > 0.05) and twelve-week (SMD = 0.45, 95% CI [−0.48, 1.38], *p* > 0.05) durations showed no effects. The sensitivity analysis indicated that the results were unstable. After we excluded Mezzasalma’s two sets of data, there was no significant heterogeneity among the other studies (*p* = 0.31, *I*^2^ = 5%; Figure 7b).

#### 3.4.6. Adverse Events

One study reported that adverse events were resolved by the end of the study, and not related to the probiotics [50]. Four RCTs [50,54,57,58] reported 54 patients with adverse events, including abdominal pain, anal irritation, gastroesophageal reflux, constipation, and cramping. The meta-analysis showed that there was no significant difference in adverse events between the probiotic and placebo groups (OR = 1.57, 95% CI [0.87, 2.82], *p* > 0.05; Figure 8).

### 3.5. Publication Bias

All outcomes were constrained by the number of studies (n < 10), so we did not adopt publication bias detection.

### 3.6. Certainty Assessment

The GRADE evidence summary of certainty showed that all five outcomes were rated as low (abdominal pain, bloating, stool consistency, adverse events, and the number of fecal *Bifidobacteria* and *Lactobacilli*) or very low (QoL). The main reasons we downgraded the quality of evidence were the risk of bias, inconsistency, and imprecision. Specifically, all outcome indicators were downgraded by one level due to the limitations of those studies. Then, except for the stool consistency scores, all other outcome indicators were downgraded by one level in terms of inconsistency. Ultimately, the imprecision of the stool consistency scores, QoL scores, adverse events, and the number of fecal *Bifidobacteria* and *Lactobacilli* were all downgraded by one level (Table 2).

## 4. Discussion

### 4.1. Summary of the Main Results

Our systematic review and meta-analysis included 10 published RCTs containing 757 IBS-C patients from France, the US, India, Canada, Korea, Italy, South Africa, and China. Compared to placebos, the meta-analysis results reveal that probiotics could improve the stool consistency of IBS-C patients and increase the number of *Bifidobacteria* and *Lactobacilli* in feces, while the effectiveness of probiotics on abdominal pain, bloating, and quality of life was unclear. In the treatment duration subgroup, we found that a short treatment duration may have been more effective in improving stool consistency and increasing the number of *Bifidobacteria* and *Lactobacilli* in feces than a long duration. As for safety, probiotics were well-tolerated and there were no serious adverse events. There were no significant differences in adverse events between the probiotic and placebo groups, suggesting that these symptoms may not be closely related to the use of probiotics. The existing results suggest that probiotics may be a safe and effective treatment for IBS-C without increasing the incidence of adverse events.

The findings reveal that probiotics might improve stool consistency in IBS-C patients, similar to the conclusion reported by Yong et al. [8]. Their study primarily focused on evaluating the efficacy of probiotics in functional constipation, supporting probiotics in improving stool frequency, stool consistency, and whole-gut transit time. However, our study focused only on patients with IBS-C to verify the effectiveness and safety of probiotics in treating constipation symptoms, so only 10 studies were included. In addition, our finding that the number of *Bifidobacteria* and *Lactobacilli* in the feces of IBS-C patients increased after probiotic treatment was consistent with that of Xiong et al. This study showed that the intestinal flora of patients with IBS was unbalanced, and the changes in the intestinal flora of patients with different subtypes of IBS were not always constant. The more common outcome was reductions in probiotic bacteria such as *Lactobacillus* and *Bifidobacterium* (compared to healthy controls), resulting in the reductions in the resistance to intestinal colonization [8,12,13]. Probiotics intervention is helpful in improving intestinal flora disorders in patients with IBS. Yet, there are some limitations in representing the intestinal flora through the detection of fecal flora because fecal flora does not stay in the intestine for a long time and will transfer with intestinal peristalsis and eventually be excreted [14]. Thus, the composition of fecal flora cannot be completely relied upon to determine the distribution of intestinal flora. A more meaningful feature would be intestinal mucosal flora, but the detection of mucosal flora is invasive and difficult to achieve in human trials. The defect in this trial method limits the credibility of our research results, which should be considered in future trial designs.

We discovered some statistical heterogeneity in the meta-analysis of abdominal pain, QoL, and bloating. The subgroup analysis results show that the effect of different treatment durations was different. Abdominal pain and QoL improved after eight weeks of intervention, while relief from bloating was required after twelve weeks. Sensitivity analysis showed that the outcomes were unstable (abdominal pain, quality of life, and bloating), so they should be interpreted cautiously. In the abdominal pain outcomes, we consider that Gayathri’s study [58] administered standard IBS therapy (antispasmodic dicyclomine) for two weeks along with a placebo twice daily for eight weeks during the trial, which might be a major source of heterogeneity in the analysis. Heterogeneity may also be caused by differences in small sample sizes (resulting in a lower ability to detect effects in individual studies) and outcome measurements. The heterogeneity reported in the QoL outcomes may be due to differences in efficacy due to the use of different types of strains or combinations of strains. One study intervened with a mixture of two probiotics at the same dose, which could lead to bias that affected the evidence for the results [53]. Currently, each probiotic strain differs in efficacy, and the effectiveness of probiotic interventions may be strain-specific. Pooling all studies using a given species may obscure the beneficial effects of other strains within that species. Our analysis needed to include more evaluable trials to systematically assess the efficacy of different strains. Additionally, the efficacy of specific probiotics in different subtypes of IBS should be focused on since the subsets of patients with IBS are important for clinicians to optimize treatment. In addition, the accuracy and extrapolation of the results could be improved by limiting the patient subgroups in trials to improve the homogeneity of the subjects when designing trials [59,60]. The use of different scales may also lead to heterogeneity as specific scale indices and measurement criteria are inherently different. The QoL scales focus on different dimensions and specific items. Moreover, most of the included trials presented an unclear risk of bias in the areas of selection bias and blinded outcome assessment, and pooling data from studies using inappropriate methods could lead to the overestimation of probiotic efficacy. Thus, future experimental designs should be standardized to improve research quality and the reliability of the results.

### 4.2. Assessment of Evidence Quality

Due to the risk of bias in the included studies, our results should be interpreted with caution. The three study designs identified to have a low-risk bias met all standards (computer-generated random numbers, double blinding, and the complete collection and reporting of data). Critically, the remaining studies were deemed to have an ambiguous determination of bias risk due to the unclear reporting of methodological issues. It was mainly manifested in the incomplete information of random sequence generation and allocation concealment and the unclear design and implementation process of the blinding method, which may have affected the authenticity of the reported results. Therefore, it is recommended that future clinical trials follow the RCT design principles and implementation guidelines and use Consolidated Standards of Reporting Trials (CONSORT) as the RCT reporting standard to improve the study quality [61,62]. Moreover, the Cochrane quality assessment tool should be used by the investigator to self-examine and improve the study design and hypothesis to improve the scientific quality and reliability of the RCT [62].

The GRADE results indicate that the overall quality of the evidence was low or very low due to methodological inconsistencies, data imprecision, and study limitations. The potential heterogeneity between the included trials may be the main source of the inconsistencies. This heterogeneity is reflected in the differences in implementation strategies and measurements of outcome indicators. Moreover, due to the small sample size of some RCTs, the total sample size did not reach the optimal information size. Further, it may have led to excessively wide confidence intervals, resulting in inaccurate effect sizes (imprecision). Finally, because most of the included RCTs were judged to have an unclear risk of bias, these methodological limitations led to the consideration of downgrading in the evidence assessment process.

### 4.3. Strengths and Limitations

The first strength is the applicability of our research results. Although inconsistent with the relevant systematic review research object, this study and other relevant reviews always show beneficial effects. We also performed a comprehensive review of the literature in an effort to reduce publication bias. A rigorous approach was adopted to ensure data accuracy and limit the impact of serious study design flaws by using standardized templates for independent data extraction and quality assessment by two reviewers. Due to the use of different scales, relevant data could not be directly synthesized. Therefore, the SMD method was adopted for the meta-analysis to minimize deviations. Subgroup analysis was performed to try to assess the effect of treatment by treatment duration. Heterogeneity was reduced by the sensitivity analysis of the high heterogeneity of some results. We also evaluated the quality of the evidence for the critical outcomes of IBS-C using the GRADE method to determine the reliability of the results. Adverse events were systematically collected and reported, and the safety of probiotics was assessed. We were particularly focused on the effect of probiotics on fecal flora changes in patients with IBS-C. Our study supplements and reinforces the evidence for IBS with constipation.

There were inevitable limitations to this study. First, despite our comprehensive search strategy, the current study only covered reports published in English, which may have led to potential language bias. Second, the results of this study are only based on the current published literature. We will continue to follow up and update the results when relevant studies emerge. In addition, since most studies only reported the effects of multiple probiotic mixtures, we were not able to analyze specific strains.

### 4.4. Clinical Implications

Our results indicate that probiotics may have beneficial effects on IBS-C patients with a certain level of safety, so it is worth considering for clinicians. Nonetheless, the small sample size and low quality make it difficult to draw definitive conclusions. Probiotics may be a valuable approach for improving stool consistency and altering the number of *Bifidobacteria* and *Lactobacilli* in feces, and short-term treatment was more effective in this analysis. The improvement in stool consistency directly alleviated the difficulty of defecating in patients with constipation. Simultaneously, the increase in fecal *Bifidobacterium* and *Lactobacillus* indicated that imbalances in the intestinal flora may improve and maintain normal gastrointestinal function. Furthermore, normal intestinal flora cannot only alleviate and improve the symptoms of IBS but also help to improve the symptoms of other gastrointestinal diseases. Consequently, probiotic therapy can be used as a complementary therapy.

## 5. Conclusions

In summary, probiotics as a safe treatment may be beneficial for improving IBS-C stool consistency and increasing the number of fecal *Bifidobacteria* and *Lactobacilli*. Given the quality of the included studies, this evidence should be used with caution. More high-quality studies with large sample sizes need to be conducted to verify the findings of this meta-analysis. Meanwhile, future research should pay particular attention to the main effects on the strain or combination of IBS, duration, IBS subtypes, individual symptoms, and patient quality of life. Using standardized symptom measurement tools, obtaining information on which strains are most effective and which subtypes of IBS-C patients might benefit most from probiotic treatment can be used to determine the number and combination of strains for ideal probiotic supplements.

## Figures and Tables

**Figure 1 nutrients-14-02482-f001:**
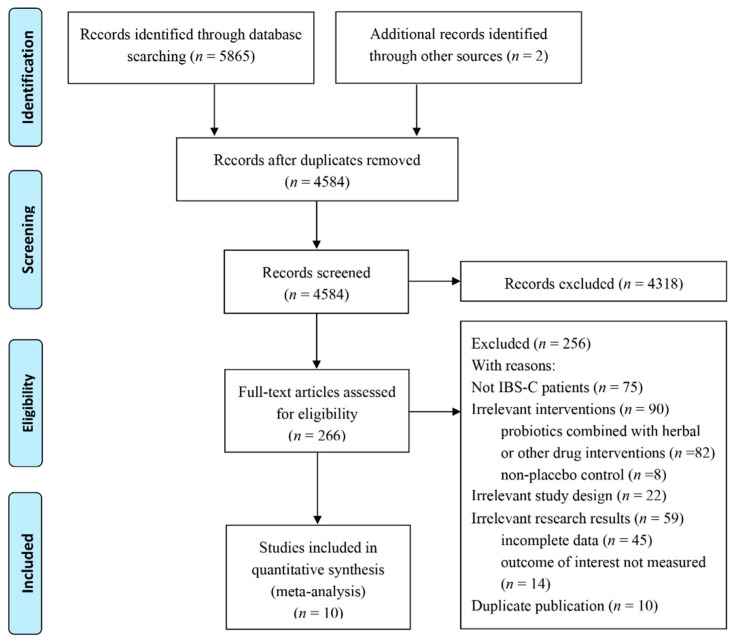
Flow diagram of the literature screening process and results.

**Figure 2 nutrients-14-02482-f002:**
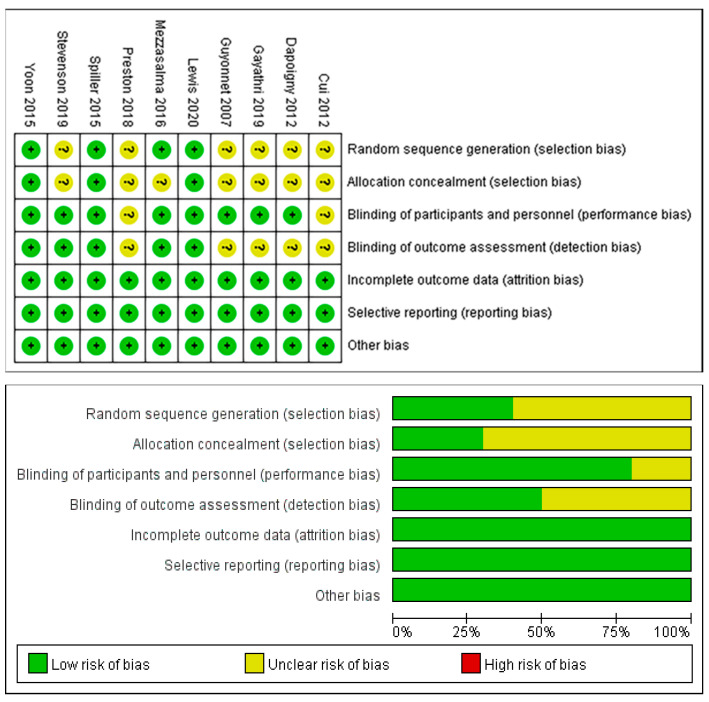
Risk of bias assessment for the included studies [49,50,51,52,53,54,55,56,57,58].

**Figure 3 nutrients-14-02482-f003:**
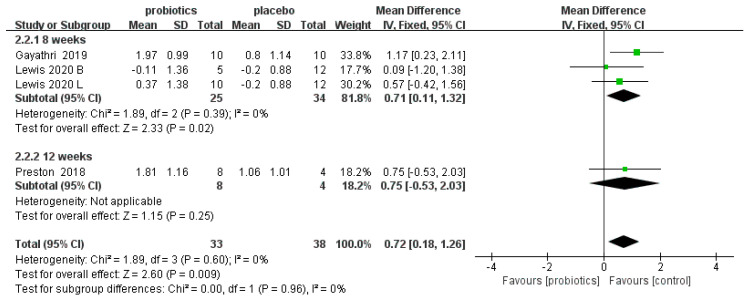
Effect of probiotics on stool consistency scores in IBS-C patients. The blue block represents the effect sizes of individual studies, black diamond block represents the combined effect size.

**Figure 4 nutrients-14-02482-f004:**
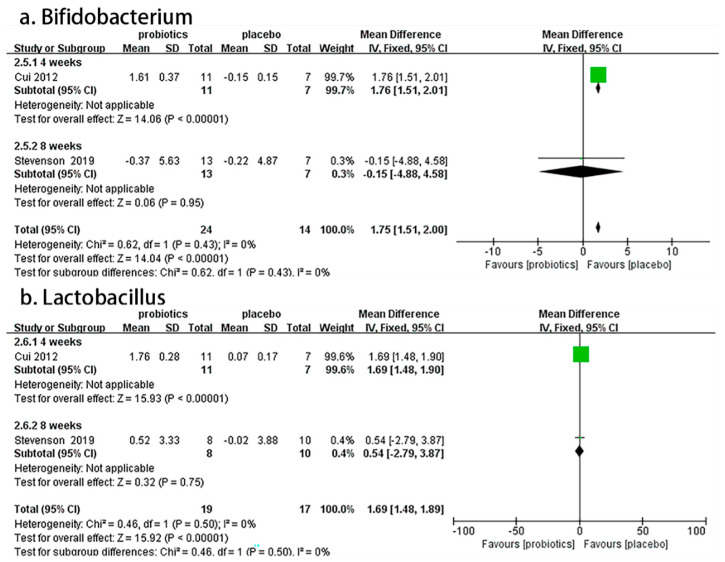
Effect of probiotics on changes in fecal counts of Bifidobacterium and Lactobacillus in IBS-C patients. The green block represents the effect sizes of individual studies, black diamond block represents the combined effect size.

**Figure 5 nutrients-14-02482-f005:**
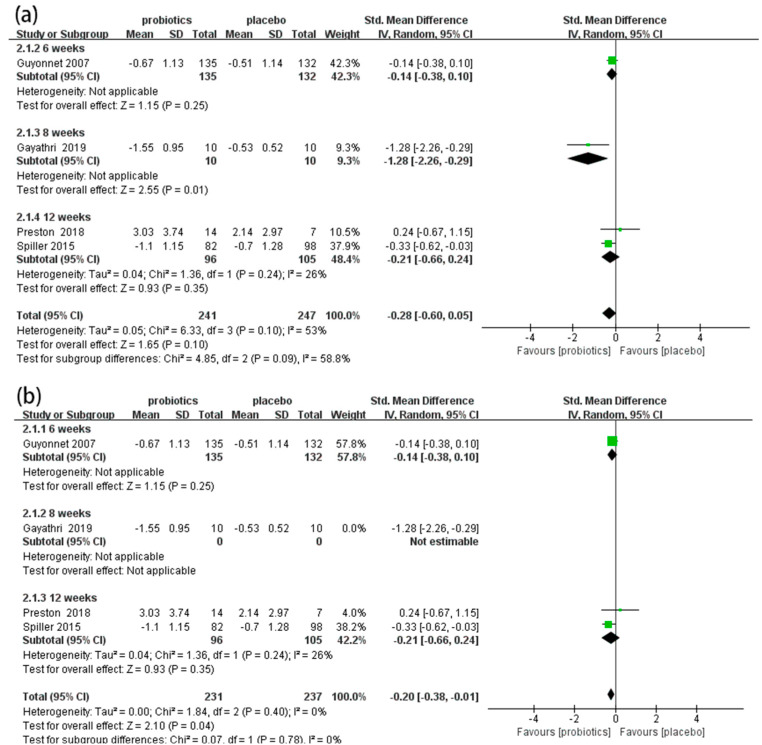
Effect of probiotics on abdominal pain scores in IBS-C patients. (**a**) subgroup analysis for abdominal pain scores; (**b**) sensitivity analysis for abdominal pain scores. The green block represents the effect sizes of individual studies, black diamond block represents the combined effect size.

**Figure 6 nutrients-14-02482-f006:**
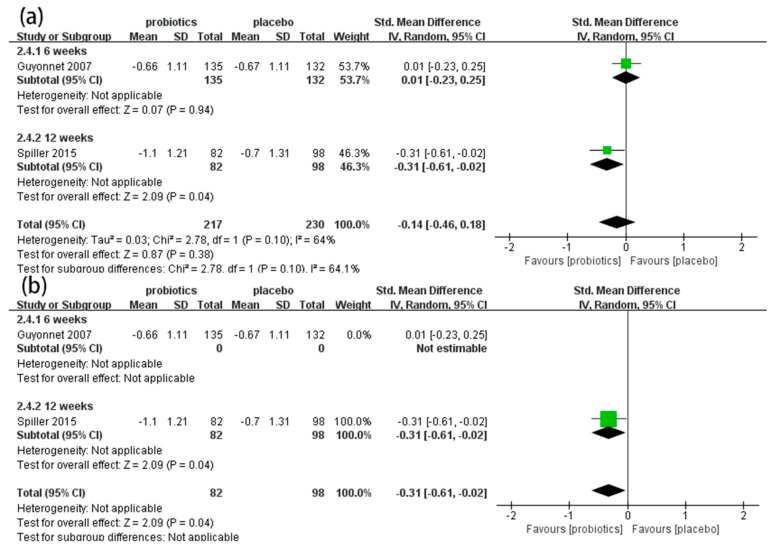
Effect of probiotics on bloating scores in IBS-C patients. (**a**) subgroup analysis for bloating scores; (**b**) sensitivity analysis bloating scores. The green block represents the effect sizes of individual studies, black diamond block represents the combined effect size.

**Figure 7 nutrients-14-02482-f007:**
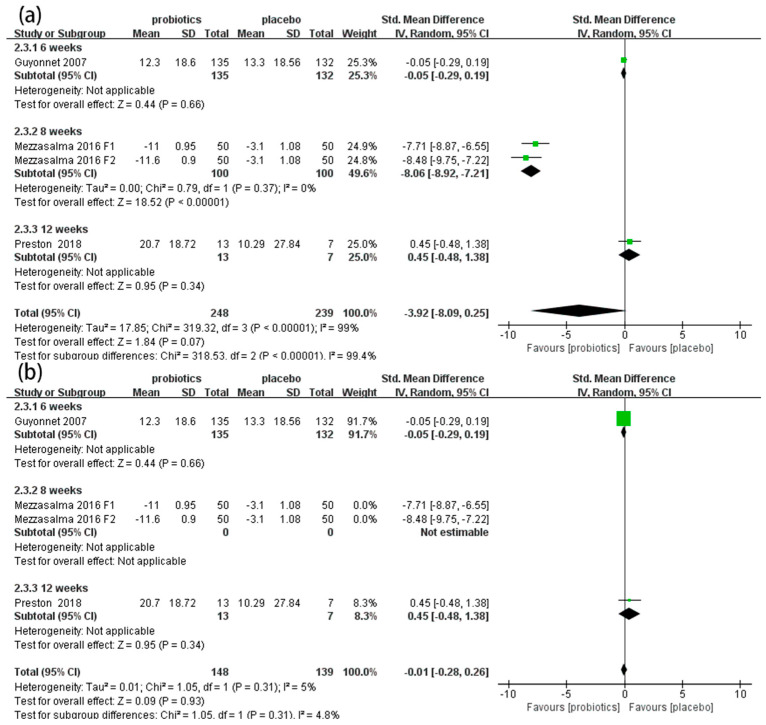
Effect of probiotics on quality of life (QoL) scores in IBS-C patients. (**a**) subgroup analysis for QoL scores; (**b**) sensitivity analysis for QoL scores. The green block represents the effect sizes of individual studies, black diamond block represents the combined effect size.

**Figure 8 nutrients-14-02482-f008:**
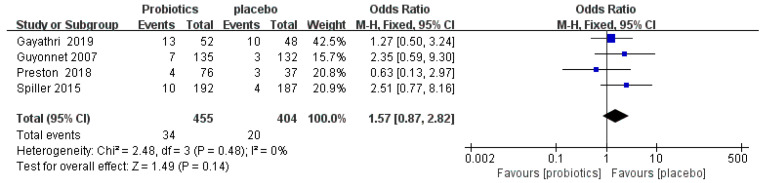
Effect of probiotics on incidence of adverse events in IBS-C patients. The green block represents the effect sizes of individual studies, black diamond block represents the combined effect size.

**Table 1 nutrients-14-02482-t001:** The essential characteristics of the included studies.

Author (Year)	Country	Sample Size (I/C)	Age (I/C)	Gender (M/F)	IBS-C Sample Size (I/C)	Diagnostic Criteria	Probiotics Genus, Strain, and Species	Dosage and Form	Type of Control	Duration	Outcomes
Yoon 2015 [49]	Korea	81 (39/42)	-	-	15 (9/6)	Rome II	*B. bifidum*, *B. lactis*, *B. longum*, *L. acidophilus*, *L. rhamnosus*, *Strept. thermophilus*	5 × 10^9^ viable cells in a lyophilized powder; 2 capsules/day	Placebo powder	4 weeks	IBS symptom relief, stool form and frequency
Spiller 2015 [57]	France	379 (192/187)	43.1 ± 15.5 45.4 ± 14.1	18/162	180 (82/98)	Rome III	*S. cerevisiae I-3856*	8 × 10^9^ cfu/g; 2 capsules/day	Calcium phosphate and maltodextrin	12 weeks	Abdominal pain, bloating, flatulence, difficulty with defecation, adverse events
Preston 2018 [50]	USA	113 (76/37)	-	-	40 *	Rome III	*L. acidophilus CL1285*, *L. casei LBC80R*, *L. rhamnosus CLR2*	50 × 10^9^ cfu; 2 capsules/day	Placebo capsule (inert ingredients)	12 weeks	Abdominal pain, distention, QoL, stool consistency and frequency, adverse events
Gayathri 2019 [58]	India	100 (52/48)	-	-	24 (12/12)	Rome III	*S. cerevisiae CNCM I-3856*	2 × 10^9^ cfu; 2 capsules/day	Placebo capsule	8 weeks	Abdominal pain, stool consistency, adverse events
Lewis 2020 [51]	Canada	285 (190/95)	-	-	28 (15/13)	Rome III	*B. longum*; *L. paracasei*	10 × 10^9^ cfu; 1 capsule/day	Placebo capsule (potato starch and magnesium stearate)	8 weeks	Stool consistency, adverse events
Dapoigny 2012 [52]	France	50 (25/25)	-	-	11 (4/7)	Rome III	*L. casei rhamnosus*	6 × 10^8^ cfu; 3 capsules/day	Placebo capsule	4 weeks	IBS severity
Mezzasalma 2016 [53]	Italy	150 (100/50) ^Δ^	I_1_:36.0 ± 11.9 I_2_:38 ± 12.1 C:38.1 ± 13.5	-	150 (100/50)	Rome III	*F1: L. acidophilus*, *L. reuteri*; *F2: L. plantarum*, *L. rhamnosus*, *B. animalis subsp. Lactis*	5 × 10^9^ cfu; 1 capsule/day	Placebo capsule	60 day	IBS-related symptom, stool consistency and frequency, QoL
Guyonnet 2007 [54]	France	267 (135/132) ^Δ^	49.4 ± 11.4 49.2 ± 11.4	68/199	267 (135/132)	Rome II	*B. animalis DN-*; *Strept. thermophilus*; *L. bulgaricus*	1.25 × 10^10^ cfu/pot, 1.2 × 10^9^ cfu/pot; 2 pots of fermented milk/day	Heat-treated yoghurt (nonliving bacteria)	6 weeks	Abdominal pain, QoL, bloating, adverse events
Stevenson 2021 [55]	South Africa	52 (35/17)	51.5 ± 9.9 49.4 ± 13.9	-	24 (16/8)	Rome II	*L. plantarum 299V*	5 × 10^9^ cfu, 1 capsule/day	Placebo capsule (micro-crystalline cellulose powder)	8 weeks	Fecal counts of *Bifidobacterium* and *Lactobacillus*
Cui 2012 [56]	China	60 (37/23)	-	-	18 (11/7)	Rome II	*B. longum*, *L. acidophilus*	6 capsules/day	Placebo capsule	4 weeks	Fecal counts of *Bifidobacterium* and *Lactobacillus*

I: intervention; C: control; M: male; F: female; CFU/d: colony-forming units per day; IBS-C: constipation-predominant irritable bowel syndrome; RI/II/III, Rome definition of IBS; B: Bifidobacterium; L.: Lactobacil-lus; S.: Saccharomyces; Strept.: Streptococcus; QoL: quality of life; *: the number of patients in each outcome is different; ^Δ^: only for constipation-predominant patients.

**Table 2 nutrients-14-02482-t002:** Summary of findings.

Estimates of Effects, Confidence Intervals, and Certainty of the Evidence for Probiotics in IBS-C Patients						
Patient or Population: IBS-C Patients Interventions: Probiotics Comparator: Placebo Setting: Hospital or No Hospital						
Outcomes	Anticipated Absolute Effects * (95% CI)		Relative Effect (95% CI)	No. of Patients (Studies)	Certainty Of Evidence (GRADE)	Comments
	Risk with Placebo	Risk with Probiotics				
Abdominal pain	-	SMD **0.28 lower** (0.60 lower to 0.05 higher)	-	488 (4 RCTs)	⨁⨁◯◯ Low ^a^	There was no difference in abdominal pain in IBS-C patients treated with probiotics compared with placebo. The evidence is uncertain because randomization, allocation concealment, and blinding were inadequately reported in most of the trials; some heterogeneity (*I*^2^ was 53%).
Stool consistency	-	MD **0.72 higher** (0.18 higher to 1.26 higher)	-	71 (3 RCTs)	⨁⨁◯◯ Low ^b^	Probiotics could improve stool consistency scores of IBS-C patients despite study limitations (lacked sufficient details on random sequence generation and allocation concealment), and sample sizes were small (imprecision).
Quality of life	-	SMD **3.92 lower** (8.09 lower to 0.25 higher)	-	487 (3 RCTs)	⨁◯◯◯ Very Low ^c^	There was no difference in QoL in IBS-C patients treated with probiotics compared with placebo. The evidence is very uncertain because randomization and allocation concealment were inadequately reported; blinding was unclear; significant heterogeneity was found (*I*^2^ was 99%); and wide confidence intervals existed (lack of precision).
Bloating	-	SMD **0.14 lower** (0.46 lower to 0.18 higher)	-	447 (2 RCTs)	⨁⨁◯◯ Low ^a^	There was no difference in bloating in IBS-C patients treated with probiotics compared with placebo. The evidence is uncertain because there was no adequate explanation for random sequence generation, allocation concealment, or blinding; significant heterogeneity(*I*^2^ was 64%).
The number of Bifidobacteria in feces	-	MD **1.75 higher** (1.51 higher to 2.00 higher)	-	38 (2 RCTs)	⨁⨁◯◯ Low ^b^	Probiotics significantly increased the number of fecal *Bifidobacteria* despite study limitations (lacked sufficient details on random sequence generation and allocation concealment), and sample sizes were small(imprecision).
The number of Lactobacilli in feces	-	MD **1.69 higher** (1.48 higher to 1.89 higher)	-	36 (2 RCTs)	⨁⨁◯◯ Low ^b^	Probiotics significantly increased the number of fecal *Lactobacilli* despite study limitations (lacked sufficient details on random sequence generation and allocation concealment), and sample sizes were small(imprecision).
Adverse events	50 per 1000	**76 per 1000** (43 to128)	OR 1.57 (0.87 to 2.82)	859 (4 RCTs)	⨁⨁◯◯ Low ^b^	There was no difference in adverse events in IBS-C patients treated with probiotics compared with placebo. The evidence is uncertain due to study limitations (randomization, allocation concealment, and blinding were inadequately reported), and sample sizes were small (imprecision) in most trials.

CI: confidence interval; RCT: randomized controlled trial; SMD: standardized mean difference; MD: standardized mean difference; *: The corresponding risk (and its 95% confidence interval) is based on the relative effect of the intervention (and its 95% CI). **GRADE Working Group grades of evidence**: high certainty (⨁⨁⨁⨁): we are very confident that the true effect lies close to that of the estimate of the effect. Moderate certainty (⨁⨁⨁◯): we are moderately confident in the effect estimate; the true effect is likely to be close to the estimate of the effect, but there is a possibility that it is substantially different. Low certainty (⨁⨁◯◯): our confidence in the effect estimate is limited; the true effect may be substantially different from the estimate of the effect. Very low certainty (⨁◯◯◯): we have very little confidence in the effect estimate; the true effect is likely to be substantially different from the estimate of effect. ^a^ We downgraded the quality to very low due to: serious study limitations and inconsistency. ^b^ We downgraded the quality to very low due to: serious study limitations and imprecision. ^c^ We downgraded the quality to very low due to: serious study limitations, imprecision, and heterogeneity. Certainty of the evidence expressed in the table by means of “⨁”and “◯” figures (⨁◯◯◯very low; ⨁⨁◯◯Low).

## Data Availability

Not applicable.

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
