# Peer review of "Effectiveness and Safety of Probiotics for Patients with Constipation-Predominant Irritable Bowel Syndrome: A Systematic Review and Meta-Analysis of 10 Randomized Controlled Trials"

_nutrients, 2022, doi:10.3390/nu14122482_

Round 1
Reviewer 1 Report
This systematic review and meta-analysis aimed to investigate the effectiveness and safety of probiotics on IBS-C patients.
The conclusions were:
probiotics as a safe treatment may be beneficial for improving IBS-C stool consistency and increasing the amount of Bifidobacteria and Lactobacillus in feces. Given the quality of the study, this evidence should be used with caution.
The topic of this systematic review and meta-analysis is very interesting. Recently another systematic review and meta-analysis was published (your reference n 8), although it is unclear in the previous meta-analysis if they were patients with IBS predominant constipation as stated in the title or functional constipation as expressed in their PICOS and summary of the study.
In my opinion, several issues need to be addressed:
1) did you include all RCT study designs (for example parallel and crossover?)
2) a large number of studies were assessed for eligibility however only 10 were included. I found that the listed reasons for exclusion both in the flow diagram of the literature screening and in the text are unclear for example irrelevant intervention or irrelevant research results. Please, could you better explain this in the text?
3) the primary outcomes included stool consistency and changes in the number of fecal colonies. The secondary outcomes were abdominal pain, bloating, QoL, and adverse events. I suggest organizing the results first primary aim and, then secondary outcomes.
4) in the Results: Characteristics of the included studies section. I think describing the age of the participants could add value to the manuscript. Why did not perform further sub-grouped analysis, for example, the duration of intervention, gender, and age?
5) the reference n 55 Stevenson C, Blaauw R, Fredericks E, Visser J, Roux S. Probiotic effect and dietary correlations on fecal microbiota profiles in irritable bowel syndrome. is incomplete and the final version should be Cheryl Stevenson, Renée Blaauw, Ernst Fredericks, Janicke Visser & Saartjie Roux (2021) Probiotic effect and dietary correlations on fecal microbiota profiles in irritable bowel syndrome, South African Journal of Clinical Nutrition, 34:3, 84-89, DOI: 10.1080/16070658.2019.1697038. Then, the year of publication in Table 1 is incorrect.
6) in the discussion “Similarly, Yong et al.[8] focused primarily on evaluating the efficacy of probiotics in functional constipation, supporting probiotics to improve stool frequency, stool consistency, and whole gut transit time. Differently, our study focused only on patients with IBS-C to verify the effectiveness and safety of probiotics in treating constipation symptoms”. There is a discrepancy between the text and the reference n 8 (Wen Y, Li J, Long Q, Yue CC, He B, Tang XG. The efficacy and safety of probiotics for patients with constipation-predominant irritable bowel syndrome: A systematic review and meta-analysis based on seventeen randomized controlled trials. International journal of surgery (London, England). 2020;79:111 119) and the reference 8 is correct.
7) it was impossible for me to retrieve the reference Xiong W, He D, Hao P. The analysis of intestinal microflora in patients with irritable bowel syndrome and the effect of probiotics. HENAN MEDICAL R ESEARCH. 2019;28(21):3859 3862. I think that at least a link should be provided to the Readers.
Reviewer 2 Report
A well written manuscript and a good contribution. I have no suggestions for improvement.
Reviewer 3 Report
The meta analysis Effectiveness and safety of probiotics for patients with constipation-predominant irritable bowel syndrome: a systematic review and meta-analysis of 10 randomized controlled trials summarizes probiotics as a treatment for IBS-C stool consistency and increasing amount of Bifidobacteria and Lactobacillus in feces.
The author stated that more high quality studies with larger samples are needed to verify this result.
I have major issues with this manuscript.
The data is not clearly presented and the novelty is not very high. It is very difficult to read trough the manuscript and the author needs to present the data more clear.
Round 2
Reviewer 1 Report
The authors addressed all issues in the point-by-point letter and, in my opinion, this systematic review with meta-analysis is improved. However, there are further concerns.
6) in the discussion the Authors said: “ In the abdominal pain outcomes, we suspect that Gayathri's study 58 administered standard IBS therapy (antispasmodic dicyclomine) for two weeks during the trial, which may affect the overall efficacy.”
I believe that this sentence casts a shadow over Gayathri's study. They described their study design in Material and methods. “The subjects were divided into two groups (Fig. 1). Subjects in group 1 (n = 48) received standard treatment for IBS for 2 weeks along with a placebo twice daily for 8 weeks. Group 2 (n = 52) received standard treatment for 2 weeks along with Saccharomyces cerevisiae CNCM I-3856 (2 × 109 c.f.u) capsules twice daily for 8 weeks.
Standard treatment for IBS included antidiarrheal—loperamide 2 mg BD 2 weeks for diarrhea-predominant IBS, antispasmodic—dicyclomine 20 mg QID for 2 weeks for constipation-predominant IBS, and for IBS-M subjects, treatment depends on the patient’s presentation (antidiarrheal/laxative)”.
Your Comment should be rephrased.
7) I retrieved the reference Xiong W, He D, Hao P. The analysis of intestinal microflora in patients with irritable bowel syndrome and the effect of probiotics. HENAN MEDICAL R ESEARCH. 2019;28(21):3859 3862. However, it was only in the Chinese language, also the abstract. There is an English version?
Author Response
6) in the discussion the Authors said: “In the abdominal pain outcomes, we suspect that Gayathri's study 58 administered standard IBS therapy (antispasmodic dicyclomine) for two weeks during the trial, which may affect the overall efficacy.”
I believe that this sentence casts a shadow over Gayathri's study. They described their study design in Material and methods. “The subjects were divided into two groups (Fig. 1). Subjects in group 1 (n = 48) received standard treatment for IBS for 2 weeks along with a placebo twice daily for 8 weeks. Group 2 (n = 52) received standard treatment for 2 weeks along with Saccharomyces cerevisiae CNCM I-3856 (2 × 109 c.f.u) capsules twice daily for 8 weeks.
Standard treatment for IBS included antidiarrheal—loperamide 2 mg BD 2 weeks for diarrhea-predominant IBS, antispasmodic—dicyclomine 20 mg QID for 2 weeks for constipation-predominant IBS, and for IBS-M subjects, treatment depends on the patient’s presentation (antidiarrheal/laxative)”.
Your Comment should be rephrased.
Response: Thank you very much for your reminder. We have rephrased our description as “In the abdominal pain outcomes, we consider that Gayathri's study58 administered standard IBS therapy (antispasmodic dicyclomine) for two weeks along with a placebo twice daily for 8 weeks during the trial, which might be a major source of heterogeneity in the analysis.”(Line 14-17, “4.1. Summary of main results”, Discussion, P15)
7) I retrieved the reference Xiong W, He D, Hao P. The analysis of intestinal microflora in patients with irritable bowel syndrome and the effect of probiotics. HENAN MEDICAL R ESEARCH. 2019;28(21):3859 3862. However, it was only in the Chinese language, also the abstract. There is an English version?
Response: Thank you very much. Since this reference was only written in Chinese, we are sorry that you cannot read the English version, so we found a newly published reference to replace this one. The research results also showed that the levels of Lactobacillus and Bifidobacterium in IBS patients were lower than those in healthy people.
The alternative reference is:
- Ji M, Huang H, Lan X. Correlation between Intestinal Microflora in Irritable Bowel Syndrome and Severity. Dis Markers. 2022;2022:1031844.

Reviewer 3 Report
After the revision there was an improvement of the manuscript and the author addressed all the comments.
Author Response
After the revision there was an improvement of the manuscript and the author addressed all the comments.
Response: Thank you very much for your review and comments.
